# Plant Disease Detection Using Deep Convolutional Neural Network

**J. Arun Pandian** [1] [ID], **V. Dhilip Kumar** [1,*] [ID], **Oana Geman** [2] [ID], **Mihaela Hnatiuc** [3] [ID], **Muhammad Arif** [4] and **K. Kanchanadevi** [1]

[1] Computer Science & Engineering, Vel Tech Rangarajan Dr. Sagunthala R&D Institute of Science and Technology, Chennai 600062, India; jarunpandian@veltech.edu.in (J.A.P.); kanchanadevicse@veltech.edu.in (K.K.)
[2] Neuroaesthetic Lab, Bioinstrumentation and Medical Tecniques Group, Faculty of Electrical Engineering and Computer Science, Stefan cel Mare University of Suceava, 720229 Suceava, Romania; oana.geman@usm.ro
[3] Electromechanical Faculty, Department of Telecomunication and Electronics, Maritime University of Constanta, 900663 Constanta, Romania; mihaela.hnatiuc@cmu-edu.eu
[4] Department of Computer Science and Information Technology, University of Lahore, Lahore 54590, Pakistan; muhammad.arif@cs.uol.edu.pk
* Correspondence: vdhilipkumar@veltech.edu.in

**Abstract:** In this research, we proposed a novel 14-layered deep convolutional neural network (14-DCNN) to detect plant leaf diseases using leaf images. A new dataset was created using various open datasets. Data augmentation techniques were used to balance the individual class sizes of the dataset. Three image augmentation techniques were used: basic image manipulation (BIM), deep convolutional generative adversarial network (DCGAN) and neural style transfer (NST). The dataset consists of 147,500 images of 58 different healthy and diseased plant leaf classes and one no-leaf class. The proposed DCNN model was trained in the multi-graphics processing units (MGPUs) environment for 1000 epochs. The random search with the coarse-to-fine searching technique was used to select the most suitable hyperparameter values to improve the training performance of the proposed DCNN model. On the 8850 test images, the proposed DCNN model achieved 99.9655% overall classification accuracy, 99.7999% weighted average precision, 99.7966% weighted average recall, and 99.7968% weighted average F1 score. Additionally, the overall performance of the proposed DCNN model was better than the existing transfer learning approaches.

**Keywords:** deep convolutional neural networks; generative adversarial network; basic image manipulation; random search; hyperparameter optimization; neural style transfer

## 1. Introduction

The diagnosis and treatment of disease are essential to improving the growth and yield of agricultural plants. For instance, an average estimated yield loss by corn crop diseases in the United States and Ontario from 2012 to 2015 was USD 76.51 per acre [1]. Manual monitoring of plant diseases will not give accurate outcomes regularly [2]. Additionally, finding domain experts for monitoring plant diseases is highly difficult and expensive for farmers. For that reason, an intelligent plant disease diagnosis system was essential to monitor the agricultural fields regularly [3]. At present, several plant disease detection methods are proposed for automatic plant disease detection using artificial intelligence techniques with fewer human efforts [4]. Deep convolutional neural network (DCNN) is a most successful image classification technique [5]. The DCNN comprises various layers, such as convolutional, pooling and fully connected layers for learning features from the training data [6].

Transfer learning uses the pre-trained neural network from one task to a similar new task. Transfer learning techniques can minimize the time for model design and training. The standard transfer learning techniques in image classification are AlexNet, DenseNet, VGG16, Inception-v3, MobileNet and ResNet [7]. The DCNN requires more data for an

efficient training process [8]. Data augmentation techniques are producing new images for the existing dataset using several data transformation techniques [9,10]. Basic image manipulation (BIM), deep convolutional generative adversarial network (DCGAN) and neural style transfer (NST) are popular data augmentation techniques [11]. The most common BIM techniques are affine transformation, scaling, cropping, flipping, padding, rotation, translation, brightness, contrast, saturation and hue. The DCGAN is an unsupervised neural network to create a new set of realistic images from the training data [12]. The GAN comprises two DCNNs, such as generator DCNN and discriminator DCNN. The generator DCNN creates new images similar to the training data; also, the discriminator network classifies the original and newly created images by the generator DCNN. DCGAN is one of the most successful image augmentation techniques in medical image processing applications [13]. The NST is an image transformation technique to produce new images using content and style reference images [12].

Additionally, the DCNNs need suitable hyperparameter values to improve the classification performance. Hyperparameters are the most significant training parameters that can influence the performance of deep learning techniques. Activation function, dropout value, epochs, filter size, learning rate, loss function and mini-batch size are the most common hyperparameters in DCNNs. The selection of the suitable value of hyperparameters is a challenging task in solving a deep learning problem. Hyperparameter tuning techniques are used to discover the most suitable hyperparameter values for the DCNNs [14]. Grid search and random search are the most popular hyperparameter tuning approaches in deep learning. High-performance computing power is needed to train the deep learning algorithm with better efficiency and less training time [15].

This article proposed a novel DCNN model for diagnosing 42 leaf diseases in 16 different plant species. This research used data augmentation and hyperparameter optimization techniques to improve the performance of the disease detection model. To produce augmented leaf images, BIM, DCGAN and NST techniques were used. There are 58 diseased and healthy plant leaf classes that were used to train the DCNN model. Random searching with the coarse-to-fine technique was used to optimize the value of the most common hyperparameters. Finally, the performance of the proposed DCNN was compared with the standard transfer learning approaches, such as AlexNet, Inception-v3-Net, ResNet-50 and VGG16Net. The rest of the article is organized as follows: Section 2 provides a detailed survey on plant leaf disease detection using artificial intelligence techniques. Section 3 describes the implementation process of the proposed DCNN model for plant leaf disease detection. Section 4 demonstrates the experimental results and related discussions of the research. Finally, Section 5 provides the conclusions and future directions of the research.

## 2. Related Works

The early detection of plant diseases is a significant step in the disease prevention and treatment process [6]. Accurate disease detection techniques can be used by the farmers for applying the prevention and treatment procedures [10,16]. The most recent plant disease detection techniques are reviewed in this section. In Ref. [4], the authors proposed a support vector machine (SVM) model for detecting sugar crop diseases. They used the hyperspectral images as an input of the SVM model for disease detection. The average classification accuracy of the hyperspectral image-based SVM technique was 78% in testing data. In Ref. [17], the authors proposed two huanglongbing disease identification models for citrus plants. The SVM and artificial neural network (ANN) were used to design the huanglongbing detection techniques. The classification accuracies of the SVM and ANN models in the test data were 92.8% and 92.2% respectively. Identification of tomato yellow leaf curl disease was achieved using the SVM technique with the quadratic kernel function proposed by the author in [18]. The overall classification accuracy of this algorithm was 92% in tomato yellow leaf curl disease detection.

Additionally, the authors in [19] discussed and compared numerous image processing and feature extraction techniques to identify the various plant diseases using their leaf

images. The authors in [20] developed a DCNN model to detect legume plant species using vein morphological patterns. In Ref. [21], the authors proposed a region-based and single-shot multi-box detector CNN model for designing the plant disease and pest identification model. The VGGNet and ResNet were used to improve the classification performance of the model. Table 1 compares the various state-of-the-art DCNN models for plant disease detection proposed by different articles.

**Table 1.** Comparison of different DCNN architecture.

| Article | Year | Specie | Number of Classes | Number of Images | Architecture | Accuracy (%) |
|---------|------|--------|-------------------|------------------|--------------|--------------|
| [22] | 2017 | Maize | 2 | 1796 | Custom | 96.7 |
| [23] | 2017 | Wheat | 3 | 3500 | Custom | 81.04 |
| [24] | 2015 | Cucumber | 3 | 800 | Custom | 94.9 |
| [25] | 2016 | Apple | 5 | 1450 | AlexNet | 97.3 |
| [26] | 2018 | Tomato | 7 | 13,262 | VGG16Net | 97.29 |
| [27] | 2018 | Maize | 9 | 3060 | GoogLeNet | 98.9 |
| [28] | 2017 | Tomato | 9 | 14,828 | GoogLeNet | 99.18 |
| [29] | 2017 | Rice | 10 | 500 | AlexNet | 95.48 |
| [2] | 2016 | Multiple | 15 | 4483 | CaffeNet | 96.3 |
| [30] | 2016 | Multiple | 38 | 54,306 | GoogLeNet | 99.35 |
| [7] | 2018 | Multiple | 38 | 54,323 | InceptionV3Net | 99.76 |
| [3] | 2019 | Multiple | 39 | 61,486 | Custom | 96.46 |
| [8] | 2018 | Multiple | 58 | 87,848 | VGG16Net | 99.53 |
| [31] | 2019 | Multiple | 79 | 46,409 | GoogLeNet | 86.5 |
| [15] | 2020 | Tomato | 10 | 18,160 | Custom | 98.7 |
| [14] | 2021 | Tomato | 10 | 3000 | Custom | 98.49 |
| [5] | 2021 | Tomato | 10 | 18,345 | AlexNet | 98.0 |
| [1] | 2022 | Multiple | 38 | 240,000 | Custom | 98.41 |

The data augmentation technique improves the diversity of training data without collecting new data. The authors in [11] compared the advantages of various data augmentation techniques in the DCNNs training process. The augmentation techniques are GAN, flipping, cropping, shifting, principal component analysis (PCA), color, noise and rotation. The result shows that the training performance of the cropping, flipping, GAN and rotation are higher than the other augmentation techniques. Additionally, the result proves that the combination of different augmentation techniques can give better performance than individuals. In Ref. [12], the authors introduced the BIM, GAN and NST augmentation techniques for plant leaf disease classification and compared the performance of each technique. The experimental result shows that the performance of the combined augmentation technique was better than the individual techniques. The authors in [32], discussed the advantages of the existing data augmentation techniques in deep learning applications. In Ref. [33], the authors proposed a DCNN for pest detection using GAN based image augmented dataset. The testing result shows that the classification performance of the GAN-based image augmented dataset was better than the non-augmented dataset. The authors in [13] discussed the advantages of the GAN augmentation technique in DCNN development. Moreover, the authors in [34] discussed the importance of hyperparameter tuning to achieve a better performance of DCNNs. The detailed survey indicates the importance of dataset size, augmentation techniques and selection of hyperparameter values in the plant leaf disease detection model. The following section presents the information about the proposed dataset and DCNN model for detecting various plant diseases from leaf images.

## 3. Materials and Methods

This section provides a complete description of the architecture and training process of the proposed DCNN model with the experimental setup and dataset preparation. The proposed plant leaf disease detection model pipeline starts with dataset preparation and

ends with model prediction. Python 3.7 programming language and TensorFlow 2.9.1, numpy Version 1.19. 2, matplotlib Version 3.5.2 and OpenCV Version 4.5.5 libraries are used for dataset preparation and DCNN model implementation. Data preparation, preprocessing, model designing and prediction tasks are performed using an HP Z240 workstation with an Intel Core i7 CPU and sixteen gigabytes of random access memory. The training and testing process of the proposed DCNN and existing state-of-the-art techniques were performed using an NVidia DGX-1 deep learning server station. The deep learning server includes two Intel Xeon E5-2694 Version 4 CPUs and eight Tesla P100 GPUs for accelerating the training process of deep neural networks. In subsequent subsections, each phase of the proposed plant leaf disease detection pipeline are discussed in detail. First, the details of the dataset preparation and preprocess are discussed in the next subsection.

### 3.1. Dataset Preparation and Preprocessing

Diseased and healthy leaf images of various plants were collected from different standard open data repositories [9,35–38]. Sixteen different plant species were used to create the plant leaf disease dataset. Each plant contains healthy and the most common disease classes in the dataset. There are 58 different classes of plant leaves, and 1 no-leaves class is present in the dataset. The collected original dataset contains 61,459 plant leaves and no-leaves images. The list of plant names with the healthy and disease classes of the proposed dataset is shown in Table 2.

**Table 2.** List of classes in the proposed dataset.

| S. No | Plant Name | Class Names |
|:---:|:---:|:---:|
| 1 | | Healthy |
| 2 | Aloe Vera | Leaf Rot |
| 3 | | Leaf Rust |
| 4 | | Healthy |
| 5 | Apple | Leaf Scab |
| 6 | | Black Rot |
| 7 | | Leaf Rust |
| 8 | | Healthy |
| 9 | Banana | Bacterial Wilt |
| 10 | | Black Sigatoka |
| 11 | Cherry | Healthy |
| 12 | | Powdery Mildew |
| 13 | | Healthy |
| 14 | | Black Spot |
| 15 | Citrus | Canker |
| 16 | | Greening |
| 17 | | Melanose |
| 18 | | Healthy |
| 19 | Corn | Common Rust |
| 20 | | Leaf Spot |
| 21 | | Northern Leaf Blight |
| 22 | | Healthy |
| 23 | Coffee | Cercospora Leaf Spot |
| 24 | | Leaf Rust |
| 25 | | Red Spider Mite |
| 26 | | Healthy |
| 27 | Grape | Black Measles |
| 28 | | Black Rot |
| 29 | | Leaf Blight |

**Table 2.** *Cont.*

| S. No | Plant Name | Class Names |
|---|---|---|
| 30 | Paddy | Healthy |
| 31 | | Brown Spot |
| 32 | | Hispa |
| 33 | | Leaf Blast |
| 34 | Peach | Healthy |
| 35 | | Bacterial Spot |
| 36 | Pepper | Healthy |
| 37 | | Bacterial Spot |
| 38 | Potato | Healthy |
| 39 | | Early Blight |
| 40 | | Late Blight |
| 41 | Strawberry | Healthy |
| 42 | | Leaf Scorch |
| 43 | Tea | Healthy |
| 44 | | Leaf Blight |
| 45 | | Red Leaf Spot |
| 46 | | Red Scab |
| 47 | Tomato | Healthy |
| 48 | | Bacterial Spot |
| 49 | | Early Blight |
| 50 | | Late Blight |
| 51 | | Leaf Mold |
| 52 | | Leaf Spot |
| 53 | | Spider Mite |
| 54 | | Target Spot |
| 55 | | Mosaic Virus |
| 56 | | Yellow Leaf Curl Virus |
| 57 | Wheat | Healthy |
| 58 | | Leaf Rust |
| 59 | no-leaves | no-leaves |

To create an even number of images in each class, data augmentation techniques were introduced. Additionally, the data augmentation techniques can increase the dataset size and reduce the overfitting during the training process of the model by adding some augmented images to the training dataset. The BIM, DCGAN and NST augmentation techniques were used to produce the augmented images in the dataset. The BIM augmentation techniques consist of image cropping, flipping, PCA color augmentation, rotation and scaling. The PCA color augmentation technique alters the intensity of the color channels using the principal component of the pixels [11]. Additionally, the image cropping, flipping, rotation and scaling techniques create augmented images by changing the color and position of the input images. There are 36,541 augmented images created by the BIM augmentation technique in the dataset.

DCGANs create augmented images that resemble the training data. The DCGAN consists of two DCNN networks, such as generator DCNN and discriminator DCNN. The generator DCNN network takes a vector of random noise and up-samples it to the training data. On the other hand, the discriminator DCNN learns to classify the real and generated images [33]. The DCGAN network was trained in the graphics processing units with training epochs of 10,000 and a mini-batch size of 64. There are 32,000 augmented images created by the DCGAN augmentation technique in the dataset. NST is another image generation technique using deep learning techniques. A modified VGG19 network was used to develop the NST augmentation model in this research. The NST model was trained with 5000 epochs on the deep learning server system. The NST models require two

different input images to generate an augmented output image, such as the content image and the style reference image. At first, the content image contains the essential features to be added to the output image. Second, the style reference image contains style patterns to apply to the output image. To generate the output image, the NST augmentation model applies the style features of the style image to the content image. The NST augmentation technique creates 17,500 augmented images in the dataset. Finally, The BIM, DCGAN and NST techniques were created for the augmented images to balance the data counts in each class of the dataset. The proposed dataset is named the PlantDisease59 dataset. These augmentation techniques increased the number of images in the dataset from 61,459 to 147,500 images. Additionally, the size of individual classes increased to 2500 images in each. Leaf images on the PlantDisease59 dataset were captured in the face-up direction. Figure 1 shows the sample augmented images generated by the BIM, DCGAN and NST techniques.

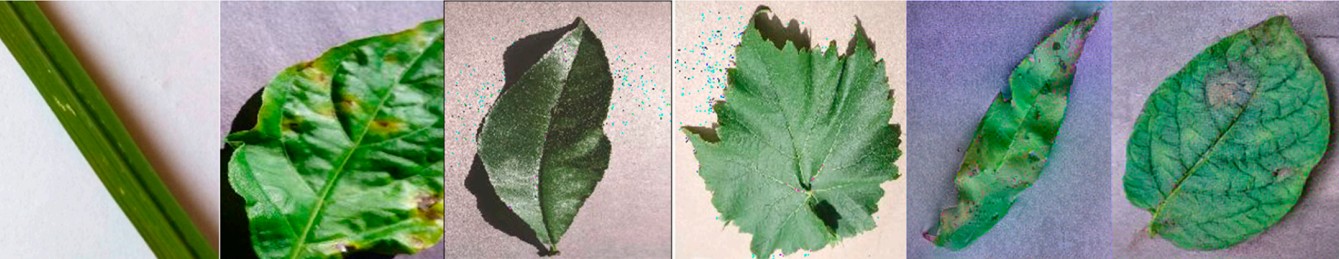

**Figure 1.** Sample augmented images using BIM, DCGAN and NST techniques.

The first two images in Figure 1 were created using BIM techniques. The third and fourth leaf images in Figure 1 were created using the DCGAN augmentation technique. The last two sample images in Figure 1 were generated using the NST technique. The random selection technique was used to select the images for training, validation and testing from the PlantDisease59 dataset. Table 3 illustrates the number of images in the training, validation and testing dataset.

**Table 3.** Training, validation and test dataset size.

| Dataset Name | Number of Images | Number of Images in Each Class |
|:---:|:---:|:---:|
| Training Set | 132,750 | 2250 |
| Validation Set | 5900 | 100 |
| Testing Set | 8850 | 150 |

The design and development of the Proposed DCNN model for plant leaf disease detection using the hyperparameter tuning techniques and the PlantDisease59 dataset are discussed in the following subsection.

### 3.2. Model Design

In this section, a DCNN model for diagnosing plant leaf diseases using the Plant-Disease59 dataset is proposed. Several DCNN models with different numbers and sizes of convolutional (Conv) and pooling layers were developed, and their performance is compared. The number of Conv layers varies from three to eight in different DCNN models. At maximum, the 14-layered deep convolutional neural network (14-DCNN) gives better training performance than other developed models. Five convolutional and five max-pooling layers were used to develop the proposed 14-DCNN model. The input images of the 14-DCNN are given into the first two-dimensional Conv layer. The dimension of the Conv layer output can be calculated using Equation (1):

$$Dimension(\text{Conv}(n,k)) = \left( \left[ \frac{n_w - f_w}{s} + 1 \right], \left[ \frac{n_h - f_h}{s} + 1 \right], f_C \right) \tag{1}$$

The input width ($n_w$) and height ($n_h$) of the first convolutional layer are 128 and 128 respectively. Additionally, the $f_w$, $f_h$ and $f_c$ represent the width, height and channels of the kernel filter of the convolutional layer. The stride ($S$) value of this Conv layer is one. The first max-pooling layer was introduced to reduce the dimension of the first Conv layer output from 126, 126, 4 to 63, 63, 4 values. The dimension of the max-pooling layer output was calculated using Equation (2):

$$Dimension(Pooling(n,k)) = \left( \left[ \frac{n_w - f_w}{s} + 1 \right], \left[ \frac{n_h - f_h}{s} + 1 \right], n_C \right) \qquad (2)$$

The $n_w$, $n_h$ and $n_c$ represent the width, height and channels of input $n$, respectively. Additionally, the $f_w$, $f_h$ and $n_c$ represent the width, height and channels of the filter ($f$) in the max-pooling layer. The output of the first max-pooling layer is given as an input of the second Conv layer.

Likewise, the second Conv layer uses the filter sizes of 16, 3, 3 values to extract the features from the data. The output size of the second Conv layer is 61, 61, 16 values. The second max-pooling layer reduces the output data size of the second Conv layer from 61, 61, 16 to 30, 30, 16 with the filter size of 2, 2 values. The third Conv layer was introduced after the second max-pooling layer. It extracts more features from the input data using the 32, 3, 3 size kernel and produces the 28, 28, 32 sized output data. The third max-pooling layer was used to reduce the size of the output data from the third Conv layer with 2, 2 sized kernels. It reduces the size of the data to 14, 14, 32 values from 28, 28, 32 values. Additionally, the fourth Conv layer uses the 64, 3, 3 sized kernels to extract the additional features from the third pooled data and generates the output data with the size of 12, 12, 64 values from the input size of 14, 14, 32 values. In addition, the fourth max pooling layer has a filter size of 2, 2 matrices and stride value of 1 step. The fourth max-pooling layer reduces the size from 12, 12, 64 values to 6, 6, 64 values. The fifth Conv layer was introduced with a filter size of 128, 3, 3 matrices. The size of the input matrix is 6, 6, 64 and the output size of the fifth Conv layer is 4, 4, 128 matrices. After the fifth Conv layer, the fifth pooing layer was used with max-pooling function and stride value of 1 step. The input size of the fifth max-pooling layer is 4, 4, 128 values, and the output size is 2, 2, 128 values. It uses the single-step stride values to apply the kernel to the input data from the fifth Conv layer. The *ReLu* activation function was used on all the above Conv layers. The *ReLu* activation function was performed by using Equation (3).

$$ReLu(x) = Max(0,\ x) \qquad (3)$$

Moreover, the flatten layer was introduced after the fifth convolutional and pooling layer. It reduces the three-dimensional data to one-dimensional data for a traditional neural network approach. The flatten layer converts the output of the fifth max-pooling layer from 2, 2, 128 values to 512 values. The first dense layer was introduced after the flatten layer in the 14-DCNN. The first dense layer increases input value from 512 to 2048 values. Equation (4) represents the individual neuron output ($z_j$) of the first dense layer. The $i$ represents the number of inputs of the first dense layer, and it ranges from 1 to 512. Additionally, the $j$ denotes the number of outputs of the layer its range from 1 to 2048 values.

$$z_j = ReLu\left( 0,\ \sum_{i}^{512} b_j + x_i w_i \right) \qquad (4)$$

The $x_i$ and $w_i$ represent the value and weight of the $i$th input of the $j$th output. Additionally, the $b_j$ denotes the bias value of the $j$th node. The dropout layer was used between the first dense and second dense layer in the 14-DCNN to avoid the overfitting issue. The second dense layer was initiated after the first dense layer and dropout layer. Similarly, the number of inputs of the second dense layer is 2048 and the output values of the neural network is 59 values. Additionally, this layer uses the softmax activation function to classify

the plant leaves. The softmax function ($\sigma$) value of $i$th neuron of the dense layer can be calculated using Equation (5).

$$softmax(\sigma(z_i)) = \frac{e^{z_i}}{\sum_{j=1}^{59} e^{z_j}} \tag{5}$$

The output class of the input image can be discovered using Equation (6).

$$Output\ Class\ (z_{out}) = max(z_1, z_2, \dots z_{59}) \tag{6}$$

This output value from $z_1$ to $z_{59}$ represents the number of diseases and healthy plant leaf and non-leaves classes in the PlantDisease59 dataset. The total number of training parameters is 5,424,583 in the 14-DCNN model. The layered structure of the 14-DCNN model is shown in Figure 2.

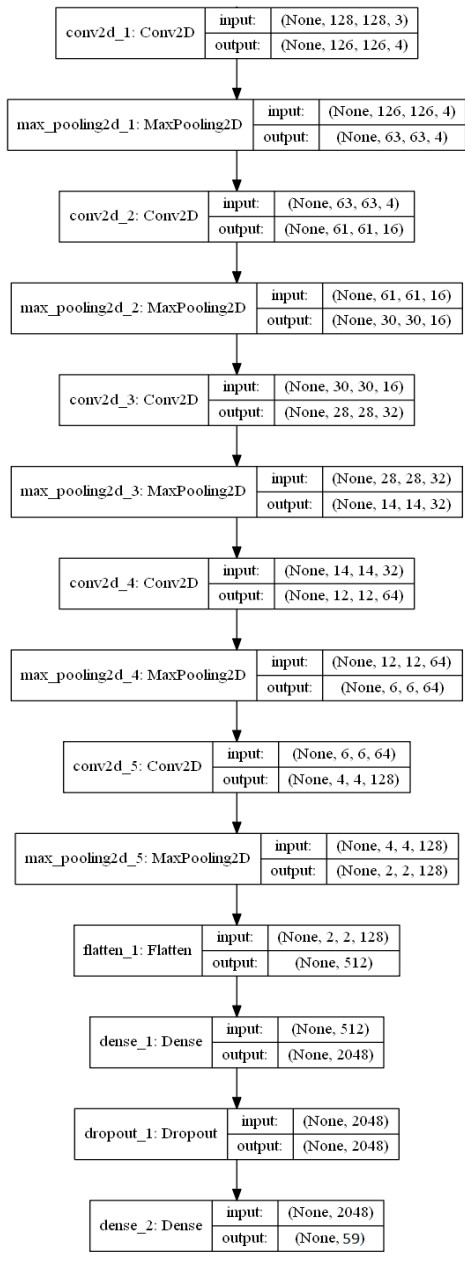

**Figure 2.** Layered structure of 14-DCNN model.

After designing the 14-DCNN model, the most suitable hyperparameter values were identified using hyperparameter tuning techniques. The random search and coarse-to-fine techniques were used to discover the suitable values of the optimizer function, mini-batch size and dropout probability of the 14-DCNN. The most common optimizers considered for the hyperparameter searching are adaptive moment estimation (Adam), stochastic gradient descent (SGD) and root mean square propagation (RMSprob). The range of the mini-batch size is between 8 and 256, incremented by 8 per value. Additionally, the dropout range varies between 0.0 and 0.5, incremented by 0.1 per value. The random searching technique chooses the random combination of hyperparameter values from the search space. The selected combination of hyperparameter values is applied to the 14-DCNN and trained with 100 epochs in parallel. From the training result, the coarse-to-fine process helps to identify the most possible hyperparameter values from the search space. Finally, the hyperparameter tuning technique discovers the SGD optimizer with a mini-batch size of 32 and the dropout probability of 0.2 gives a better performance than other values. To identify the learning rate (Lr) and momentum of the SGD optimizer, a similar random search approach was used with a possible combination of the values. Table 4 shows the most common hyperparameters of the 14-DCNN model for the plant leaf disease detection model with their values.

**Table 4.** Optimized hyperparameters of the 14-DCNN.

| Hyperparameter | Value |
|---|---|
| Batch Sizes | 32 |
| Dropout Value | 0.2 |
| Loss | Categorical Cross entropy |
| Optimizer | SGD with Lr = 0.0001 and momentum = 0.9 |
| Activation function for Conv layer | ReLu |

The random search with the coarse-to-fine technique offers significantly improved searching performance than the grid search and simple random search optimization techniques. The optimized hyperparameter values and the PlantDisease59 were used to train the proposed 14-DCNN model for diagnosing the diseases from the plant leaf images.

### 3.3. Model Training

The proposed 14-DCNN was trained using the optimized hyperparameters and augmented dataset in the deep learning server environment. The model was trained with different epoch values between 100 and 3000. The 1000 epochs gave the maximum validation accuracy and minimum loss. The training time of the proposed model with 1000 epochs was 7452 s in Nvidia DGX-1 deep learning server. Figure 3 demonstrates the training and validation performance of the proposed 14-DCNN model for the identification of plant leaf diseases.

The 14-DCNN model achieved a training accuracy of 99.993% and a validation accuracy of 99.985%. Training and validation accuracies of the proposed 14-DCNN are higher than the other proposed DCNN models. The proposed 14-DCNN took 7452 s for the training process in the MGPUs environment. The training time of the proposed 14-DCNN was smaller than the transfer learning techniques since the number of convolutional and pooling operations of the proposed 14-DCNN are lesser than the transfer learning techniques. Finally, the architecture and weights of the proposed 14-DCNN model were stored as a hierarchical data (H5) file for the further prediction process.

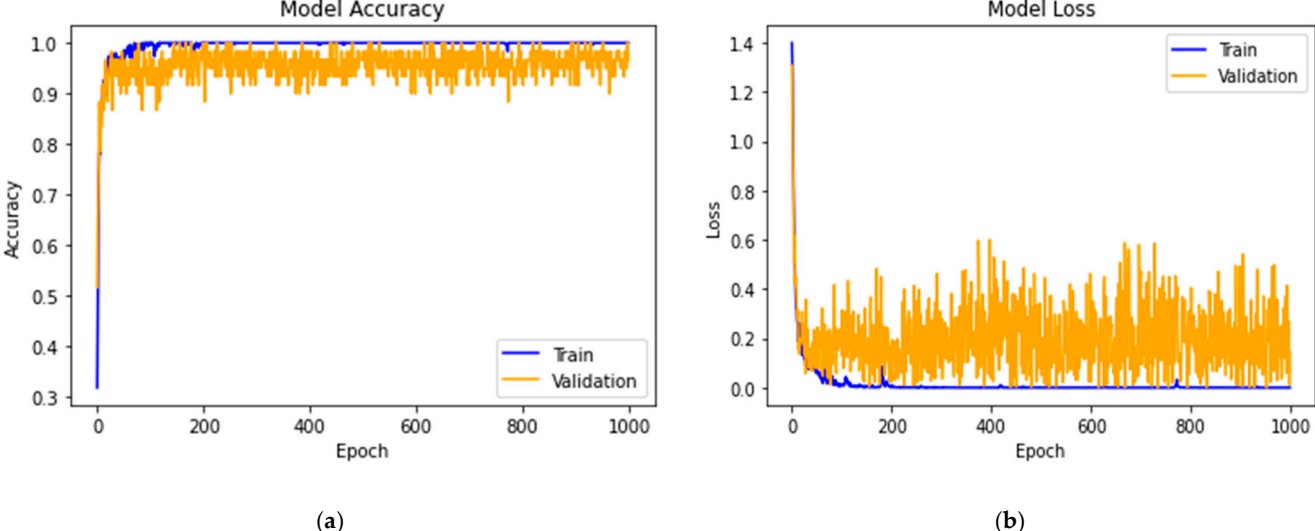

(**a**)                                                                 (**b**)

**Figure 3.** (**a**) Accuracy and (**b**) loss of the proposed 14-DCNN model.

*3.4. Model Prediction*

The saved 14-DCNN model architecture and weights were used to detect the diseases of the various plants from the input images. The real-time plant disease images were given as an input of the 14-DCNN model. The 14-DCNN model successfully predicted the plant name and disease from the input images. The matplotlib package was used to visualize the predictions of the model. Figure 4 shows the random sample prediction from the proposed 14-DCNN model.

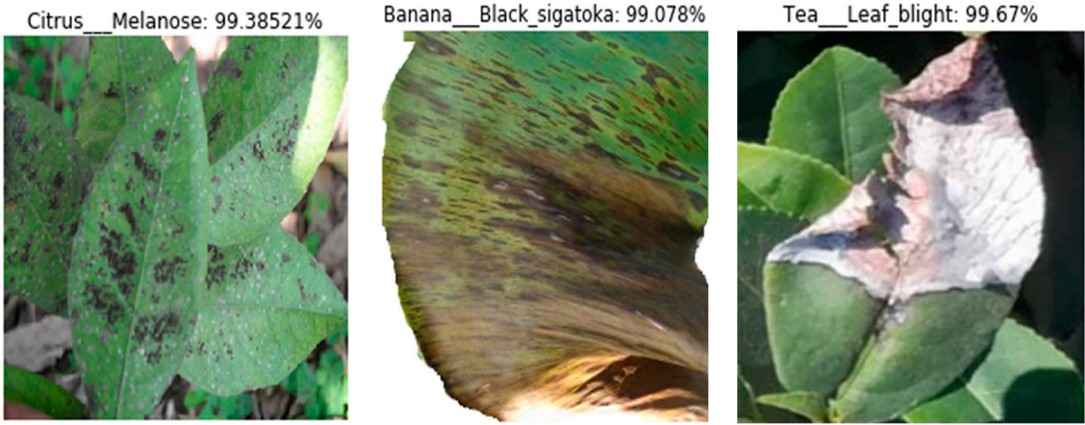

**Figure 4.** Sample predicted images using the 14-DCNN model.

Finally, the 14-DCNN model was converted as a TensorFlow lite (tflite) file using a TensorFlow lite converter with a latency optimization approach. The tflite file can be used to deploy the model in mobile and embedded devices for real-time prediction.

**4. Results**

This section examines the performance of the proposed 14-DCNN model, using various performance evaluation approaches and testing datasets. Additionally, the performance of the proposed 14-DCNN model is compared with other state-of-the-art techniques. The state-of-the-art techniques are AlexNet [25], Inception-v3-Net, ResNet-50 and VGG16Net [7,8]. These models were trained on a deep learning server system using the PlantDisease59 dataset and tested using the testing dataset. The proposed and existing mod-

els were trained only on face-up leaf images. So, the models will give the best performance on the face-up direction positioned images.

Occlusion sensitivity can visualize the most important part of the input image for classification identified by the trained model. It can measure the sensitivity of the neural network to occlusion in different regions of the data, using small perturbations of the data. This region is known as the occluding region [7]. The white and light blue color pixels of the image illustrate the most essential parts of being classified into the expected class. The dark blue color region of the image has minimum features for the classification. Occluding stride and occluding size of the selected sensitivity map are 10 and 30, respectively. Figure 5 illustrates the occlusion sensitivity of the proposed 14-DCNN model on sample test data.

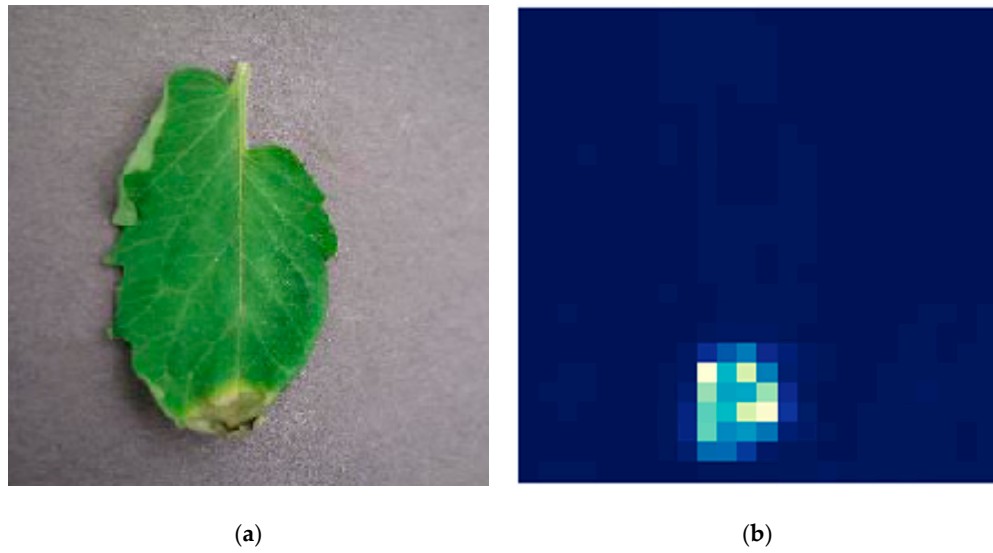

(**a**)          (**b**)

**Figure 5.** (**a**) Original image, (**b**) occlusion sensitivity of 14-DCNN.

The confusion matrix is a summary of predictions made by the classification techniques. The confusion matrix of the classification technique represents the true positive (TP), true negative (TN), false positive (FP) and false negative (FN) values of every single class [3]. The area under the receiver operating characteristic (AUC-ROC) curve is one of the popular metrics that is used to evaluate the performance of learning algorithms. The ROC curve plots the difference between the true positive rate (TPR) and false positive rate (FPR) [15]. The TPR and FPR are calculated using Equations (7) and (8).

$$\text{TPR} = \frac{\text{TP}}{\text{TP} + \text{FN}} \tag{7}$$

$$\text{FPR} = \frac{\text{FP}}{\text{FP} + \text{TN}} \tag{8}$$

The AUC-ROC curve of the cherry healthy and strawberry leaf scorch classes shows the classification advantage of the proposed 14-DCNN. Figure 6 illustrates the AUC-ROC curve of the cherry healthy and strawberry leaf scorch classes, using the proposed 14-DCNN model.

The performance of the proposed 14-DCNN model and state-of-the-art techniques is compared, using the most common performance metrics, such as classification accuracy, precision, recall and F1-score [3]. At first, the classification accuracy is defined as the correctly classified images divided by the total number of testing images. Precision is the second most important performance evaluation metric in classification techniques. Precision is defined as the number of correctly identified results divided by the number of correctly identified and correctly rejected results that are predicted by the model. Precision is used to find the correct proportion of the classification.

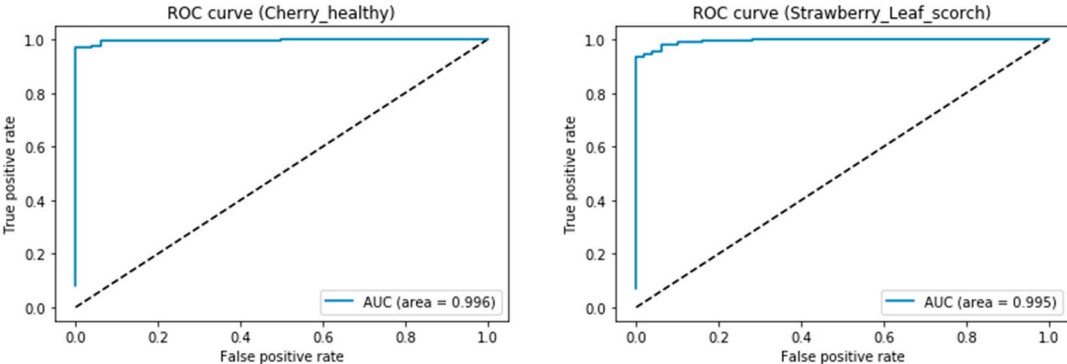

**Figure 6.** Sample AUC-ROC curves of 14-DCNN.

The third most important performance evaluation technique is recall. The recall is the number of correctly identified results divided by the number of correctly identified and incorrectly rejected results. The recalls are used to determine the proportion of actual positives that were correctly identified. The F1 score is one of the widely used metrics for the performance evaluation of machine learning algorithms. The F1 score is defined as the harmonic mean between precision and recall. The F1 score value represents the prediction advantage of the classification techniques. The following Equations (9)–(12) were used to calculate the accuracy, weighted average precision, weighted average recall and weighted average F1 score of the classification techniques.

$$\text{Classification accuracy} = \frac{\text{TP} + \text{TN}}{\text{TP} + \text{TN} + \text{FP} + \text{FN}} \tag{9}$$

$$\text{Precision} = \frac{\text{TP}}{\text{TP} + \text{FP}} \tag{10}$$

$$\text{Recall} = \frac{\text{TP}}{\text{TP} + \text{FN}} \tag{11}$$

$$\text{F1 Score} = 2 \times \frac{(\text{precision} \times \text{recall})}{(\text{precision} + \text{recall})} \tag{12}$$

The range of the classification accuracy, precision, recall and F1 score is between 0 (0%) and 1 (100%). The classification performance of the proposed 14-DCNN model on individual classes in the dataset is illustrated in Table 5.

Figure 7 compares the performance of the proposed 14-DCNN and existing state-of-the-art classification techniques using accuracy, weighted average precision, weighted average recall and weighted average F1 score.

The comparison results show the accuracy, precision, recall, and F1 scores of the proposed 14-DCNN are higher than the AlexNet, Inception-v3-Net, ResNet-50 and VGG16Net. Additionally, the complexity of the proposed 14-DCNN and other transfer learning techniques is illustrated in Table 6.

The complexity analysis result shows that the number of trainable parameters and model size of the proposed 14-DCNN model are lesser than the existing transfer learning techniques. The smaller number of trainable parameters and small model size will reduce the complexity of the model prediction process. The comparison results illustrate that the performance of the proposed 14-DCNN is higher than the AlexNet, Inception-v3-Net, ResNet-50 and VGG16Net on plant leaf disease classification.

**Table 5.** Class-wise performance of proposed 14-DCNN model.

| Plant Name | Class Names | PRECISION | RECALL | F1-SCORE |
|---|---|---|---|---|
| Aloe Vera | Healthy | 1 | 1 | 1 |
| | Leaf Rot | 1 | 0.98667 | 0.99329 |
| | Leaf Rust | 0.98684 | 1 | 0.99338 |
| Apple | Healthy | 1 | 1 | 1 |
| | Leaf Scab | 1 | 1 | 1 |
| | Black Rot | 1 | 0.98667 | 0.99329 |
| | Leaf Rust | 1 | 1 | 1 |
| Banana | Healthy | 1 | 1 | 1 |
| | Bacterial Wilt | 0.99338 | 1 | 0.99668 |
| | Black Sigatoka | 1 | 0.99333 | 0.99666 |
| Cherry | Healthy | 1 | 1 | 1 |
| | Powdery Mildew | 1 | 1 | 1 |
| Citrus | Healthy | 1 | 1 | 1 |
| | Black Spot | 0.98684 | 1 | 0.99338 |
| | Canker | 1 | 1 | 1 |
| | Greening | 1 | 1 | 1 |
| | Melanose | 1 | 0.98667 | 0.99329 |
| Corn | Healthy | 1 | 1 | 1 |
| | Common Rust | 1 | 1 | 1 |
| | Leaf Spot | 1 | 1 | 1 |
| | Northern Leaf Blight | 1 | 1 | 1 |
| Coffee | Healthy | 1 | 1 | 1 |
| | Cercospora Leaf Spot | 1 | 1 | 1 |
| | Leaf Rust | 1 | 1 | 1 |
| | Red Spider Mite | 1 | 1 | 1 |
| Grape | Healthy | 1 | 1 | 1 |
| | Black Measles | 1 | 0.98 | 0.9899 |
| | Black Rot | 0.96774 | 1 | 0.98361 |
| | Leaf Blight | 1 | 1 | 1 |
| Paddy | Healthy | 1 | 1 | 1 |
| | Brown Spot | 0.98039 | 1 | 0.9901 |
| | Hispa | 1 | 1 | 1 |
| | Leaf Blast | 1 | 1 | 1 |
| Peach | Healthy | 1 | 1 | 1 |
| | Bacterial Spot | 0.98658 | 0.98 | 0.98328 |
| Pepper | Healthy | 1 | 1 | 1 |
| | Bacterial Spot | 1 | 0.98667 | 0.99329 |
| Potato | Healthy | 1 | 1 | 1 |
| | Early Blight | 1 | 1 | 1 |
| | Late Blight | 1 | 1 | 1 |
| Strawberry | Healthy | 1 | 1 | 1 |
| | Leaf Scorch | 1 | 1 | 1 |
| Tea | Healthy | 1 | 1 | 1 |
| | Leaf Blight | 1 | 0.98667 | 0.99329 |
| | Red Leaf Spot | 0.98684 | 1 | 0.99338 |
| | Red Scab | 1 | 1 | 1 |

**Table 5.** *Cont.*

| Plant Name | Class Names | PRECISION | RECALL | F1-SCORE |
|---|---|---|---|---|
| Tomato | Healthy | 1 | 1 | 1 |
| | Bacterial Spot | 1 | 1 | 1 |
| | Early Blight | 1 | 1 | 1 |
| | Late Blight | 1 | 0.99333 | 0.99666 |
| | Leaf Mold | 0.99338 | 1 | 0.99668 |
| | Leaf Spot | 1 | 1 | 1 |
| | Spider Mite | 1 | 1 | 1 |
| | Target Spot | 1 | 1 | 1 |
| | Mosaic Virus | 1 | 1 | 1 |
| | Yellow Leaf Curl Virus | 1 | 1 | 1 |
| Wheat | Healthy | 1 | 1 | 1 |
| | Leaf Rust | 1 | 1 | 1 |
| No Leaves | No Leaves | 1 | 1 | 1 |

**Table 6.** Complexity comparison of proposed 14-DCNN and existing model.

| | AlexNet | Inception-v3-Net | ResNet-50 | VGG16Net | 14-DCNN |
|---|---|---|---|---|---|
| No. of Parameters | 44,752,739 | 24,937,283 | 26,722,211 | 39,443,043 | 17,928,571 |
| Model Size (MB) | 133 | 92 | 98 | 128 | 37 |

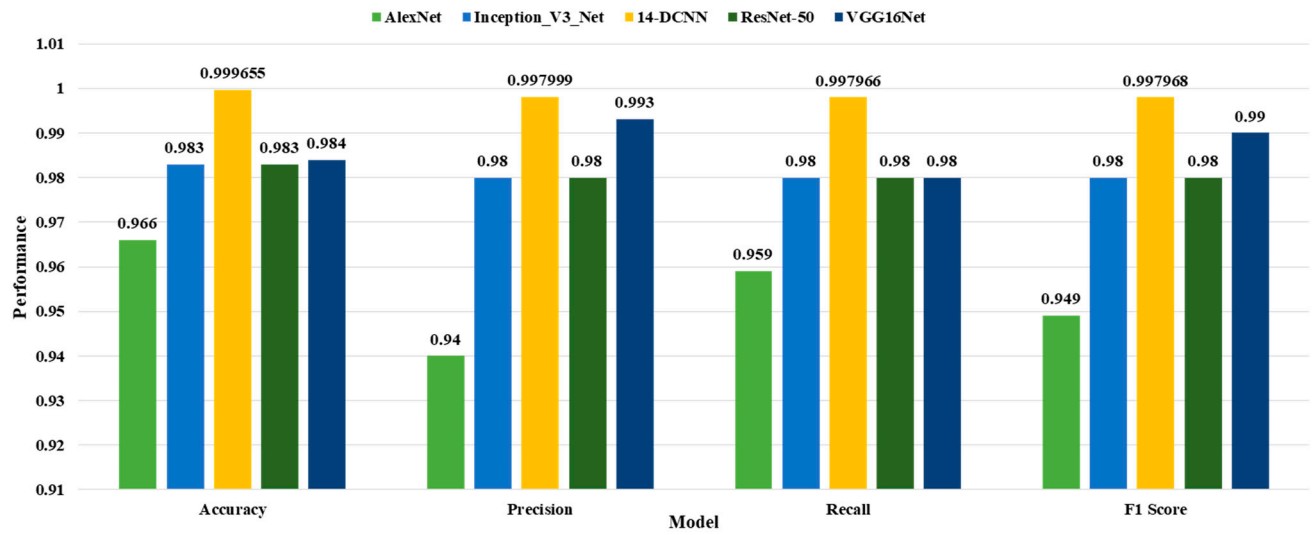

**Figure 7.** Performance comparison of proposed 14-DCNN and existing model.

## 5. Conclusions

A novel DCNN model was proposed to detect plant leaf diseases from leaf images in this research. The proposed 14-DCNN model was designed and trained to detect 42 leaf diseases in 16 plants through leaf images. The data augmentation and hyperparameter optimization techniques were also used to enhance the performance of the 14-DCNN in this research. Three augmentation techniques were used to enhance the dataset size to 147,500 images. The augmentation techniques are NST, DCGAN, and BIM. The individual class size, including original and augmented images, of the dataset, was 2500 images. The

14-DCNN comprises five Conv and five max-pooling layers. The random search with the coarse-to-fine technique was used to optimize the value of the hyperparameter for training the proposed 14-DCNN model. Training of the most successful 14-DCNN model was completed with the training and validation dataset of 139,000 images and optimized hyperparameter values. The proposed 14-DCNN model achieved a classification accuracy of 99.9655%, a precision value of 99.7999%, a recall value of 99.7966%, and an F1 score of 99.7968% on the training dataset. The optimized hyperparameter values and the data augmentation techniques had a considerable influence on the results of the proposed DCNN model. Compared with standard transfer learning techniques, the proposed 14-DCNN model has higher classification performance. An extension of this research will be adding new classes of plant diseases and an increasing number of training images in the dataset and modifying the architecture of the DCNN model using more convolutional and other layers. In the future, we plan to estimate the possibility of plant disease and analyze the severity using the deep learning technique. Moreover, we will extend disease detection from plant leaves to other parts of the plants, such as flowers, fruits, and stems.

**Author Contributions:** Conceptualization, J.A.P. and V.D.K.; writing—original draft preparation, J.A.P. and V.D.K.; writing—review, designing, analysis and editing, J.A.P., V.D.K., O.G., M.H., M.A. and K.K.; supervision, V.D.K.; funding acquisition, M.H. All the authors contributed in writing, reviewing, and structuring of the work. All authors have read and agreed to the published version of the manuscript.

**Funding:** This work was supported by a grant of the Romanian National Authority for Scientific Research and Innovation, CCCDI-UEFISCDI, project number 203, COFUND-ICT-AGRI-FOOD-MERIAVINO-1, within PNCDI III.

**Institutional Review Board Statement:** Not applicable.

**Informed Consent Statement:** Not applicable.

**Data Availability Statement:** Not applicable.

**Conflicts of Interest:** The authors declare no conflict of interest.

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
