# Peer review of "Plant Disease Detection Using Deep Convolutional Neural Network"

_applsci, doi:10.3390/app12146982_

Round 1
Reviewer 1 Report
To detect plant leaf diseases,a novel 14-layered Deep Convolutional Neural Network is proposed, where three image augmentation techniques were used. The proposed DCNN model achieves good performance on datasets.
After reviewing the paper, the comments are given as follows:
1. About title of the paper, "Fourteen Layered" should be deleted.
2. The innovation in the abstract is not prominent enough, and the logic needs to be improved.
3. Images from the PlantDisease59 dataset are captured in the face-up direction, for images shot from different perspectives, what will be the effect of the model in the paper?
4. Although the proposed method of the paper has high accuracy, its advantages over other algorithms are not obvious as shown in Figure 7. More comparative experiments with the advanced methods should be done.
Author Response
Responses to Reviewers
Dear Editors and Reviewers:
Thank you for your letter and for the reviewers’ comments concerning our manuscript ID:
applsci-1792170
Title: Plant Disease Detection Using Fourteen Layered Deep Convolutional Neural Network
Authors: J. Arun Pandian, V. Dhilip Kumar *, Oana GEMAN, HNATIUC MIHAELA, Muhammad Arif, K. Kanchanadevi
Those comments are all valuable and very helpful for revising and improving our paper, as well as the important guiding significance to our research. We have studied the comments carefully and have made correction which we hope meet with approval. All revisions have been marked in blue in this manuscript. The main corrections in the paper and the responses to the reviewers’ comments are as follows.
Reviewer -1:
- About title of the paper, "Fourteen Layered" should be deleted.
Answer: Title was revised based on the suggestion.
- The innovation in the abstract is not prominent enough, and the logic needs to be improved.
Answer: Novelty and results-related information are included in the revised abstract.
- Images from the PlantDisease59 dataset are captured in the face-up direction, for images shot from different perspectives, what will be the effect of the model in the paper?
Answer: The proposed and existing models were trained only on face-up leaf images. So, the models will give the best performance on the face-up direction positioned images. We explained the same in sections 3.1 and 4.
- Although the proposed method of the paper has high accuracy, its advantages over other algorithms are not obvious as shown in Figure 7. More comparative experiments with the advanced methods should be done.
Answer: In the revised article we included the class-wise performance, Occlusion sensitivity and AUC-ROC metrics.
Thank you for the valuable advice.
Should you have any questions, please contact us without hesitate.
Sincerely yours,
Oana GEMAN
Associate Professor, Ph.D., Habill.
Medical Bioengineer
IEEE Senior Member
Stefan cel Mare University of Suceava, Romania
Phone: +40 0754 212 277
oana.geman@usm.ro
Reviewer 2 Report
In this paper, the authors propose a simple deep learning model for plant disease detection. They also apply several data augmentation techniques such as BIM, DCGAN, and NST to increase the number of data sets. Their experimental results show that an overall classification accuracy of 99.8% was achieved. Although the research topic looks practical and interesting, some revisions are recommended to make the paper clearer and more accurate.
- Line 34: Reorder the references to start from [1].
- Line 51: Check the indentation.
- Line 93 & 95: Define the abbreviation of ANN at line 93, not at line 95.
- Line 108: In Table 1, add one column and indicate the specie of each dataset.
- Line 160: Abbreviation of Principal Component Analysis (PCA) is double defined.
- Line 190: In Figure 1, indicate which techniques are applied for each image.
- Line 207: All equation number such as (1), (2) should be placed at the end of line. Correct all equation numbers.
- Line 208 ~ 256: All variables such as nw, nh, s, zi, bj, etc. in the text should be edited by equation editor. For example, nw -> nw
- Line 212 ~ 235: 126, 126, 4 -> 126 x 126 x 4. Correct all other places too.
- Line 254 & 259: equation 5 -> Equation 5, figure 2 -> Figure 2.
- Line 295: In Figure 3, separate loss and accuracy plots. That is, draw accuracy graphs of training and validation results in one plot. Loss plot too.
- Line 298: much higher -> only about 1% higher than other models such as Inception, ResNet, and VGG16.
- Line 298: other DCNN models -> indicate the models.
- Line 369: In Figure 7, why all the metric values of 14-DCNN are same as 0.998? Check the values of other models too.
- Line 371 & 390: much higher, superior -> too exaggerated.
- Line 372: Add complexity comparison of 14-DCNN and other models.
- Line 386-387: The metric values are different in abstract, Figure 7, and here. Make the values the same.
Author Response
Responses to Reviewers
Dear Editors and Reviewers:
Thank you for your letter and for the reviewers’ comments concerning our manuscript ID:
applsci-1792170
Title: Plant Disease Detection Using Fourteen Layered Deep Convolutional Neural Network
Authors: J. Arun Pandian, V. Dhilip Kumar *, Oana GEMAN, HNATIUC MIHAELA, Muhammad Arif, K. Kanchanadevi
Those comments are all valuable and very helpful for revising and improving our paper, as well as the important guiding significance to our research. We have studied the comments carefully and have made correction which we hope meet with approval. All revisions have been marked in blue in this manuscript. The main corrections in the paper and the responses to the reviewers’ comments are as follows.
Reviewer -2:
- Line 34: Reorder the references to start from [1].
Answer: The reference list is reordered in the revised manuscript
- Line 51: Check the indentation.
Answer: Indentation of the entire document was updated.
- Line 93 & 95: Define the abbreviation of ANN at line 93, not at line 95.
Answer: Changed as per the suggestion
- Line 108: In Table 1, add one column and indicate the specie of each dataset.
Answer: We added a new column about the specie.
- Line 160: Abbreviation of Principal Component Analysis (PCA) is double defined.
Answer: Changed as per the reviewer’s suggestion.
- Line 190: In Figure 1, indicate which techniques are applied for each image.
Answer: Sample images for each technique are included.
- Line 207: All equation number such as (1), (2) should be placed at the end of line. Correct all equation numbers.
Answer: Changed as per the reviewer’s suggestion.
- Line 208 ~ 256: All variables such as nw, nh, s, zi, bj, etc. in the text should be edited by equation editor. For example, nw -> nw
8a. Line 212 ~ 235: 126, 126, 4 -> 126 x 126 x 4. Correct all other places too.
Answer: Changed as per the suggestions
- Line254&259:equation5->Equation5,figure2->Figure2.
Answer: Changed the equation5->Equation5,figure2->Figure2 in the revised manuscript
- Line 295: In Figure 3, separate loss and accuracy plots. That is, draw accuracy graphs of training and validation results in one plot. Loss plot too.
Answer: Included the separate plots for training and validation
- Line 298: much higher -> only about 1% higher than other models such as Inception, ResNet, and VGG16.
Answer: Changed as per the suggestions
- Line 298: other DCNN models -> indicate the models.
Answer: Highlighted the model details
- Line 369: In Figure 7, why all the metric values of 14-DCNN are same as 0.998? Check the values of other models too.
Answer: The values are not the same. We rounded up the values. Now, w included the actual values alone in the revised article.
- - Line 371 & 390: much higher, superior -> too exaggerated.
Answer: Changed as per the suggestions
- Line 372: Add complexity comparison of 14-DCNN and other models.
Answer: Added the complexity information in section 3.3 in the revised manuscript.
- Line 386-387: The metric values are different in abstract, Figure7, and here. Make the values the same.
Answer: We changed the typo errors in the values.
Thank you for the valuable advice.
Should you have any questions, please contact us without hesitate.
Sincerely yours,
Oana GEMAN
Associate Professor, Ph.D., Habill.
Medical Bioengineer
IEEE Senior Member
Stefan cel Mare University of Suceava, Romania
Phone: +40 0754 212 277
oana.geman@usm.ro
Round 2
Reviewer 1 Report
The quality of the paper has been significantly improved, some minor errors should be modified before the publication.
Author Response
Spelling and grammar errors are corrected with the help of native English-speaking colleagues and grammar tools.
Reviewer 2 Report
1. Redraw Figure 3 as in the example below.
2. Add a table for model complexity comparion as in the example below.
3. Reedit all variables in the text by using MS Word Equation Editor.

Author Response
- Redraw Figure 3 as in the example below.
Answer: We changed the figure as per the reviewer's suggestion
- Add a table for model complexity comparison as in the example below.
Answer: The comparison table is included as per the reviewer's suggestion
- Reedit all variables in the text by using MS Word Equation Editor.
Answer: Reedited all the variables using equation editor as per the reviewer's suggestion